# Too Committed to Switch Off—Capturing and Organizing the Full Range of Work-Related Rumination from Detachment to Overcommitment

**DOI:** 10.3390/ijerph20043573

**Published:** 2023-02-17

**Authors:** Oliver Weigelt, J. Charlotte Seidel, Lucy Erber, Johannes Wendsche, Yasemin Z. Varol, Gerald M. Weiher, Petra Gierer, Claudia Sciannimanica, Richard Janzen, Christine J. Syrek

**Affiliations:** 1Wilhelm Wundt Institute of Psychology, Leipzig University, D-04109 Leipzig, Germany; charlotte.seidel@winterlinge.eu (J.C.S.); dt40usit@studserv.uni-leipzig.de (L.E.); rj92gabu@studserv.uni-leipzig.de (R.J.); 2Federal Institute for Occupational Safety and Health, Section 3 Work and Health, D-01099 Dresden, Germany; wendsche.johannes@baua.bund.de; 3Educational Psychology, Goethe University Frankfurt, D-60629 Frankfurt, Germany; varol@psych.uni-frankfurt.de (Y.Z.V.); weiher@psych.uni-frankfurt.de (G.M.W.); 4Work and Organizational Psychology, University of Hagen, D-58084 Hagen, Germany; petra.gierer@googlemail.com (P.G.); c.sciannimanica@web.de (C.S.); 5Business Psychology, University of Applied Sciences Bonn-Rhein-Sieg, D-53359 Rheinbach, Germany; christine.syrek@h-brs.de

**Keywords:** work-related rumination, overcommitment, psychological detachment, burnout, irritation, problem-solving pondering, positive work reflection, negative work reflection, affective rumination, satisfaction with life

## Abstract

Work-related thoughts during off-job time have been studied extensively in occupational health psychology and related fields. We provide a focused review of the research on overcommitment—a component within the effort–reward imbalance model—and aim to connect this line of research to the most commonly studied aspects of work-related rumination. Drawing on this integrative review, we analyze survey data on ten facets of work-related rumination, namely (1) overcommitment, (2) psychological detachment, (3) affective rumination, (4) problem-solving pondering, (5) positive work reflection, (6) negative work reflection, (7) distraction, (8) cognitive irritation, (9) emotional irritation, and (10) inability to recover. First, we apply exploratory factor analysis to self-reported survey data from 357 employees to calibrate overcommitment items and to position overcommitment within the nomological net of work-related rumination constructs. Second, we apply confirmatory factor analysis to self-reported survey data from 388 employees to provide a more specific test of uniqueness vs. overlap among these constructs. Third, we apply relative weight analyses to assess the unique criterion-related validity of each work-related rumination facet regarding (1) physical fatigue, (2) cognitive fatigue, (3) emotional fatigue, (4) burnout, (5) psychosomatic complaints, and (6) satisfaction with life. Our results suggest that several measures of work-related rumination (e.g., overcommitment and cognitive irritation) can be used interchangeably. Emotional irritation and affective rumination emerge as the strongest unique predictors of fatigue, burnout, psychosomatic complaints, and satisfaction with life. Our study is intended to assist researchers in making informed decisions on selecting scales for their research and paves the way for integrating research on the effort–reward imbalance into work-related rumination.

## 1. Introduction

Work-related thinking during off-job time has been identified as a key contributor to prolonged occupational stress [1,2,3]. The *perseverative cognition hypothesis* posits that stressor-related thinking can be as stressful as experiencing the stressors themselves [3]. Applying the *perseverative cognition hypothesis* to occupational stress, work-related thinking may be *the* critical factor in explaining prolonged stress experiences and should be associated with risks to employee well-being [4,5]. According to the *allostatic load model* [6], prolonged stress responses may manifest in psychological states of fear, tension, or anxiety in the short term (primary allostatic load, initial adaptation) [7]. Over time, these processes at the psychological level pave the way to ill-health at the physiological level, as reflected in elevated hormonal setpoints (secondary allostatic load, set-point adjustment) and, in the long run, to the onset of diseases such as cardiovascular complaints (tertiary allostatic load, health outcomes) [7]. Given the pivotal role of work-related thinking during off-job time in explaining the deleterious effects of work, it is not surprising that a significant amount of research has been devoted to examining various aspects of work-related rumination [8,9]. 

Work-related rumination has been conceptualized in the literature [8,9] in various ways. Drawing on Martin and Tesser’s broad definition of rumination [10], Weigelt and colleagues [9] proposed the term “work-related rumination” as an overarching term for any kind of recurrent work-related thoughts during off-job time. In line with this reasoning, and by slightly modifying Martin and Tesser’s generic definition of rumination to apply to work-related thinking, we refer to work-related rumination as conscious thoughts during off-job time (e.g., work breaks, evenings, weekends, or vacations) that revolve around work and that recur in the absence of immediate environmental demands requiring such thoughts. 

Research in occupational health psychology has focused primarily on five facets of work-related rumination in a broad sense, namely (1) psychological detachment [11], (2) affective rumination [1], (3) problem-solving pondering [1], (4) positive work reflection [12], and (5) negative work reflection [13]. There is empirical evidence that these five facets are distinct factors and relate differentially to specific indicators of employee well-being, such as burnout and satisfaction with life [8,9]. However, additional constructs related to work-related rumination have been developed independently. For instance, overcommitment has been extensively studied in public health research, e.g., [14,15], but rarely considered in occupational health psychology [16]. Overcommitment is a focal variable in the effort–reward imbalance model [17,18] conceptualized in medical sociology as “a set of attitudes, behaviors, and emotions that reflect excessive striving in combination with a strong desire of being approved and esteemed” [19] (p. 55). 

A closer look at the content of focal questionnaire items suggests that there may be a considerable overlap in what is captured in work-related rumination facets and overcommitment. Work-related rumination is studied pursuing disparate lines of research within and across disciplines [9]. This state of affairs makes construct clean-up or more integrated and consistent research efforts within occupational health psychology and across disciplines challenging, if not impossible [20]. Our aim is to provide a more comprehensive understanding of how overcommitment and various facets of work-related rumination are related to employee well-being and health. 

We contribute to the literature on work-related rumination in three ways: First, based on a factor analytical approach, we integrate overcommitment—a construct commonly used in medical sociology research—with the five dominant facets of work-related rumination studied in occupational health psychology. Second, we include four less extensively studied and (partially) earlier developed constructs such as cognitive irritation and emotional irritation [21], distraction [22], and inability to recover [23,24,25], to capture the full range of frequently studied work-related rumination constructs and so to work towards an integrative framework of work-related rumination. We examine convergence (vs. distinctiveness) of constructs to quantify the overlap vs. the uniqueness of all these constructs. Third, we employ relative weight analysis and scrutinize the relevance of each facet of work-related rumination by examining the criterion-related validity with regard to key aspects of employee well-being and health, namely regarding (1) physical fatigue, (2) cognitive fatigue, (3) emotional fatigue, (4) burnout, (5) psychosomatic complaints, and (6) satisfaction with life.

### 1.1. Overcommitment

The construct of overcommitment (OVC) originates from the effort–reward imbalance model (ERI) [17]. According to the ERI model, an imbalance between the effort exerted by the employee and rewards provided by the employer constitutes a risk to individual health. Overcommitment is conceptualized as a disposition to cope with demanding situations by excessive engagement and urge to be in control [16,26]. 

A recent systematic review found overcommitment linked to impaired self-reported (e.g., fatigue, insomnia) and physiological health outcomes (e.g., blood pressure, atherogenic lipids) [26]. Moreover, overcommitment is positively linked to indicators of impaired energetic well-being, such as fatigue [27], burnout [28,29,30,31,32,33,34,35], emotional exhaustion [15,36,37,38,39,40,41,42], vital exhaustion [43,44], and need for recovery [27,45]. Other systematic reviews [26,46] found that overcommitment was associated with higher levels of (psycho)somatic complaints, also supported by more recent research [47,48,49]. However, the relationship between overcommitment and satisfaction with life has not been extensively studied; one study on students found a moderately negative relationship [50]. 

Prototypical items used to measure overcommitment focus on an individual’s inability to withdraw from work obligations (see also [17] for the broader concept of “immersion”) and being preoccupied with work-related issues during off-job time. Overcommitment has rarely been studied side by side with any facet of work-related rumination despite considerable similarities to constructs, such as psychological detachment. Consistent with this idea, we will briefly describe the most commonly applied facets of work-related rumination and review the available empirical evidence on such relationships.

### 1.2. The “Famous Five” Facets of Work-Related Rumination

The most commonly used facets of work-related rumination in occupational health psychology include (1) psychological detachment, (2) affective rumination, (3) problem-solving pondering, (4) positive work reflection, and (5) negative work reflection [8,9]. There is evidence that these “famous five” constructs capture related yet unique aspects of work-related rumination in off-job time [9]. 

*Psychological detachment*—“the sense of being away from the work situation” [11] (p. 579)—has been widely studied, and many reviews [2] and meta-analyses [51,52,53,54] have consistently found that psychological detachment relates to higher levels of employee well-being and health. Jimenez et al. [8] described psychological detachment as the construct that captures work-related thinking during off-job time in its purest form, as it neither explicitly captures the content and valence of thoughts about work, nor taps into affective experiences associated with these thoughts. 

Cropley and Zijlstra [1] have argued that being merely “switched on” in off-job time may not be problematic, but that the quality of work-related thinking does make a difference. Accordingly, they have proposed a *tripartite conceptualization of work-related rumination*, which distinguishes between detachment, affective rumination, and problem-solving pondering. Cropley and Zijlstra describe *affective rumination* as intrusive, pervasive, recurring thoughts about work which are experienced as negative in affective terms [1]. Jimenez et al. [8] have argued that affective rumination explicitly captures affective experiences accompanying work-related rumination during off-job time (and, at least implicitly, thought content and valence), and should therefore be considered a prototypical “contaminated” work-related rumination construct (i.e., a construct that captures aspects beyond the core of work-related thinking per se, e.g., concomitants of rumination). Empirical evidence has shown that detachment and affective rumination are distinct, but highly negatively correlated (see [8,55] for meta-analyses and [9] for a review). 

*Problem-solving pondering* is the third facet and is characterized as “prolonged mental scrutiny of a particular problem or an evaluation of previous work in order to see how it can be improved, but it does not involve the emotional process that sustains arousal as in affective rumination”. [1]. According to Jimenez et al. [8], problem-solving pondering is located midway on the continuum ranging from pure to contaminated. Problem-solving pondering is considered to be somewhat contaminated because typical items are explicit about the content of thoughts, but do not directly capture the affect experienced while thinking about work. 

Weigelt et al. [9] have recently argued that research on *work reflection* should be included under the work-related rumination umbrella to create a more coherent and integrated nomological network of work-related rumination. In a similar vein, Jimenez et al. [8] have conceptually integrated positive and negative work reflection in their meta-analysis of work-related rumination. *Positive work reflection* originated from recovery research and refers to thinking about the positive sides of one’s job [12]. It has been studied either as a standalone construct [56], in conjunction with psychological detachment [57] or negative work reflection [13]. According to Jimenez et al. [8], positive work reflection is to some degree contaminated as typical items are explicit about the content and valence of thoughts. Empirically, positive work reflection correlates moderately with problem-solving pondering, but has only slight associations with the other facets of work-related rumination [9]. Positive work reflection unambiguously captures unique aspects of work-related rumination and has unique associations with indicators of employee well-being [58]. 

*Negative work reflection* refers to thinking about the negative aspects of one’s job and recognizing what one does not like about it [13]. Conceptually, it is similar to affective rumination as it refers to negative ways of thinking about work during off-job time. Unlike psychological detachment, negative work reflection is explicit about the content and valence of thoughts, and is to some degree contaminated [8]. However, unlike affective rumination, negative work reflection emphasizes cognition rather than affect, and does not explicitly capture affective experiences possibly accompanying thoughts about the negative sides of one’s job [8,9]. Negative work reflection correlates moderately negatively with psychological detachment, and positively with problem-solving pondering and with affective rumination [9]. Hence, negative work reflection can be considered a distinct facet of work-related rumination with strong links to affective rumination. 

### 1.3. Less Extensively Studied Facets of Work-Related Rumination

In this section, we introduce additional aspects of work-related rumination that have been conceptualized alongside the “famous five” facets, namely cognitive irritation, emotional irritation, distraction, and inability to recover. 

Irritation refers to subjectively perceived cognitive and emotional strain in a work context. Mohr and colleagues distinguish between two aspects, *cognitive* and *emotional irritation*. *Cognitive irritation* refers to the inability to mentally switch off and was sometimes defined in terms of rumination [21,59,60]. Weigelt et al. [9] found that cognitive irritation emerges as a factor distinct from the “famous five”. However, *cognitive irritation* correlated highly with psychological detachment (*r* = −0.77) and hence can be considered largely redundant with psychological detachment. *Emotional irritation* describes the level of irritability and taps into tense activation as a reaction to goal discrepancies [21]. Hence, there is a marked similarity to affective rumination as conceptualized in the work-related rumination questionnaire (WRRQ) [22]. Consistent with this rationale, Weigelt et al. [9] found that emotional irritation is distinct from the “famous five” but highly correlated with affective rumination (*r* = 0.67). Although there seems to be a high degree of overlap between cognitive and emotional irritation and the “famous five” facets of work-related rumination, inclusion of both is warranted when studying the full range of work-related rumination. 

*Distraction* originates from the tripartite conceptualization of work-related rumination, and there is a subscale of the WRRQ labeled in the original paper as either distraction or detachment [22]. Cropley et al. [22] used the terms distraction and psychological detachment interchangeably as the overlap in content between detachment as measured in the recovery experience questionnaire (REQ; the standard measure of psychological detachment) [61] and the corresponding items of the WRRQ subscale is considerable. To avoid confusion between detachment as operationalized in the WRRQ and its operationalization in the REQ, whenever we refer to the WRRQ subscale, we consistently use the label “distraction”. We scrutinize whether the practice of using measures of distraction and psychological detachment interchangeably [9] is justified.

A final facet of work-related rumination we consider is *inability to recover*, which is one of four subscales of the faulty attitudes and behaviors analysis [24]. The FABA originates from an integration of Type A behavior research [62] and Action Regulation Theory [63] in the work context. Inability to recover refers to general attitudes towards the job. Symptoms include a chronic preoccupation with the job, high work-related effort investment, disturbed relaxation and recovery (e.g., frequent work-related thoughts and sleep problems), blurred boundaries between work and private domains, and strong feelings of work-related responsibility. Accordingly, there is a high degree of conceptual similarity and overlap between overcommitment and inability to recover [64]. Richter and colleagues [65] found that inability to recover correlated at *r* = −0.75 with psychological detachment. Hence, according to the conventions for determining convergent and discriminant validity of constructs ([66] suggest a cut-off of 0.70), inability to recover is largely redundant with psychological detachment. Including inability to recover in this study is conducive to integrating research on the FABA to overcommitment and the “famous five” facets of work-related rumination. 

### 1.4. Links between Overcommitment and Other Facets of Work-Related Rumination

According to our literature research (see Appendix A), empirical evidence linking overcommitment to work-related rumination is almost non-existent. Among the studies identified, not all reported correlations between overcommitment and the other facets of work-related rumination were considered [31,41,67], leaving us with a very small set of studies. 

Across two cross-sectional survey studies, Gillet and colleagues [45] explored recovery profiles and reported correlations between overcommitment, psychological detachment, and rumination. They applied a prototypical measure of psychological detachment and a measure of work-related rumination that does not align with any of the “famous five” facets of work-related rumination, albeit being most similar to cognitive irritation. The authors reported that overcommitment correlates strongly with psychological detachment and rumination (Psi > 0.70). This initial piece of empirical evidence is consistent with the idea that overcommitment is conceptually very close to specific facets of work-related rumination. In a multi-source survey study [68], overcommitment correlated strongly negatively with both self- and spouse-rated psychological detachment. 

Drawing on theoretical considerations on the content overlap across scales and on the evidence reviewed above, we expect that overcommitment will converge strongly with psychological detachment. Given that the correlations in the published research covered a broad range from *r* = −0.52 [68] to −0.83 [45] and given that factor analyses in prior research supported the distinction between overcommitment and psychological detachment [45], we expect that overcommitment will emerge as a factor distinct from, but highly correlated with, psychological detachment. Given the nascent state of research on the convergence vs. distinctiveness across overcommitment and the other work-related rumination facets, we formulate the following open research question: 

**Research question 1**: Is overcommitment empirically distinct from the other facets of work-related rumination? How strongly does (1) overcommitment correlate with (2) psychological detachment, (3) affective rumination, (4) problem-solving pondering, (5) positive work reflection, (6) negative work reflection, (7) distraction, (8) cognitive irritation, (9) emotional irritation, and (10) inability to recover?

From a practical perspective, it seems particularly important to identify the most “toxic” or “salubrious” facets of work-related rumination. Hence, we consider how overcommitment and the other facets of work-related rumination relate to and differentially predict well-being and health outcomes. We focus on those outcomes that have frequently been studied together with overcommitment, the “famous five” facets, and other facets of work-related rumination. Overcommitment is consistently linked to indicators of impaired energetic well-being, such as fatigue, emotional exhaustion, and burnout. Meta-analyses of psychological detachment [8,51,52,53,54] have also consistently found links to indicators of energetic well-being. We therefore focus on three aspects of work fatigue as suggested in the tripartite conceptualization of fatigue, namely (1) physical, (2) cognitive, and (3) emotional fatigue [69]. We supplement this by including (4) burnout as an alternative reflection of a suboptimal energetic resource status [70,71]. We consider links to (5) psychosomatic complaints because these are prototypical indicators of primary allostatic load [7] and have frequently been studied as correlates or consequences of overcommitment [26]. Finally, we explore the relative predictive power regarding (6) satisfaction with life—a major aspect of generic (rather than job-context-specific) well-being—because it has been frequently studied as a correlate of work-related rumination, e.g., [51]. Rather than merely comparing bivariate correlations between overcommitment and other facets of work-related rumination to these well-being and health outcomes, we explicitly provide a comparison of the unique variance explained by each facet of work-related rumination when tested concurrently with the other facets. This comparison yields a more precise understanding of the relative importance of each facet when taking into account overlap with similar constructs. Given that there is almost no empirical research comparing more than two or three facets of work-related rumination and how these are linked to well-being and health outcomes, we present no hypothesis, but formulate a research question:

**Research question 2:** Which facets of work-related rumination explain the largest portion of unique variance in (1) physical fatigue, (2) cognitive fatigue, (3) emotional fatigue, (4) burnout, (5) psychosomatic complaints, and (6) satisfaction with life?

## 2. Materials and Methods

### 2.1. Procedure

We conducted an online cross-sectional self-report survey study between June 2021 and April 2022. We advertised the study by posting the link to the survey on a website of the University of Hagen. The University of Hagen is a distance learning university with a large psychology program. Unlike typical universities, most students hold regular (full-time) jobs and study alongside these jobs. Therefore, typical distance learning students are more similar to regular employees than to students at traditional universities [72]. We offered a certificate of study participation commensurate with the duration of the survey, namely one hour. Typically, psychology students have to collect certificates for study participation equivalent to 30 h to complete the degree program. We advertised this study as a survey targeted at individuals in employment and working for at least 20 h per week.

### 2.2. Samples

The data for Sample 1 and Sample 2 were collected within a comprehensive cross-sectional self-report survey study including variables beyond the scope of this paper (e.g., a pictorial scale of energetic activation [73,74], job demands, personal resources, and other aspects of energetic well-being). After data collection, and in line with the best practices of scale development [75], we divided the total sample into two subsamples to run the exploratory analyses (Sample 1) and the confirmatory analyses (Sample 2) independently.

We describe the specific data cleaning steps and report the resulting sample sizes of remaining participants after each step in Appendix A. After data cleaning, Sample 1 consisted of 357 participants and Sample 2 consisted of 388 participants. In Sample 1, 270 participants identified themselves as female, 85 as male, and two as non-binary. Average age was 32.32 years (*SD* = 9.64, range: 19–62). The participants worked in diverse industries. Their average tenure with the current organization was 5.17 years. In Sample 1, 278 individuals did not have a leadership position. 

In Sample 2, 299 participants identified themselves as female, 88 as male, and one as non-binary. Average age was 32.13 years (*SD* = 9.54, range: 18–64). Participants came from diverse sectors. Their average organizational tenure was 5.48 years. In Sample 2, 301 individuals did not hold a managerial position. We provide a more detailed description of the two samples in Appendix A. 

### 2.3. Measures

#### 2.3.1. Overcommitment

We applied all six items of the Overcommitment subscale (OVC) of the German Effort–Reward Imbalance Questionnaire (ERIQ) by Siegrist et al. [76]. Responses in our study ranged from 1 (*strongly disagree*) to 5 (*strongly agree*). A sample item for overcommitment is “As soon as I get up in the morning I start thinking about work-related problems”.

#### 2.3.2. Psychological Detachment

We applied all four items of the Psychological Detachment subscale of the German Recovery Experience Questionnaire (REQ) by Sonnentag and Fritz [61]. Responses ranged from 1 (*strongly disagree*) to 5 (*strongly agree*). A sample item is “During time after work, I forget about work”.

#### 2.3.3. Affective Rumination

We applied all five items of the Affective Rumination subscale of the Work-Related Rumination Scale (WRRQ) by Cropley et al. [22] to measure affective rumination. The WRRQ contains 15 validated items across three subscales, namely Affective Rumination, Problem-Solving Pondering, and Distraction. The WRRQ has been adapted to German [9]. Responses ranged from 1 (*very seldom or never*) to 5 (*very often or always*). A sample item is “I become tense when I think about work-related issues during my free time”.

#### 2.3.4. Problem-Solving Pondering

We applied all five items of the Problem-Solving Pondering subscale of the German adaptation [9] of the Work-Related Rumination Questionnaire (WRRQ) by Cropley et al. [22] to measure problem-solving pondering. Responses ranged from 1 (*very seldom or never*) to 5 (*very often or always*). A sample item is “I find solutions to work-related problems in my free time”.

#### 2.3.5. Positive Work Reflection

We applied all items of the German validated 4-item questionnaire Positive Work Reflection by Binnewies et al. [13] to measure positive work reflection during leisure time. Responses ranged from 1 (*strongly disagree*) to 5 (*strongly agree*). A sample item is “During leisure time, I think about the good sides of my work”.

#### 2.3.6. Negative Work Reflection

We applied all items of the German validated 4-item questionnaire Negative Work Reflection from Binnewies et al. [13] to measure negative work reflection during leisure time. Responses ranged from 1 (*strongly disagree*) to 5 (*strongly agree*). A sample item is “During leisure time, I consider the negative aspects of my job”.

#### 2.3.7. Cognitive Irritation

We applied all three items of the Cognitive Irritation (“rumination”) subscale of the German Irritation Scale by Mohr et al. [64] to measure cognitive irritation. Responses ranged from 1 (*never/rarely*) to 5 (*very often/always*). A sample item is “Even at home I often think of my problems at work”.

#### 2.3.8. Emotional Irritation

We applied all five items of the Emotional Irritation (“irritability”) subscale of the German Irritation Scale by Mohr et al. [64] to measure emotional irritation. Responses ranged from 1 (*never/rarely*) to 5 (*very often/always*). A sample item is “I get grumpy when others approach me”.

#### 2.3.9. Distraction

We applied all five items of the Distraction subscale of the German adaptation [9] of the Work-Related Rumination Scale (WRRQ) by Cropley et al. [22] to measure distraction. Responses ranged from 1 (*very seldom or never*) to 5 (*very often or always*). A sample item is “I find it easy to unwind after work”. 

#### 2.3.10. Inability to Recover

We applied all six items of the Work Obsession/Inability to Recover subscale of the German Faulty Attitudes and Behavior Analysis Questionnaire (FABA) [24]. Responses ranged from 1 (*strongly disagree*) to 5 (*strongly agree*). A sample item is “I find it difficult to switch off after work”.

#### 2.3.11. Physical Fatigue

We applied all six items of the physical fatigue subscale of the Work Fatigue Inventory (WFI-3D) by Frone and Tidwell [69]. The WFI-3D consists of 18 items across three subscales, namely physical fatigue, mental fatigue, and emotional fatigue. The scale has been *adapted to and validated in German by Frone et al. [77]. Responses ranged from 1 (*never*) to 5 (*always*). We asked participants to refer to the past three months. A sample item is “How often did you feel physically exhausted at the end of the workday?”

#### 2.3.12. Mental Fatigue

We applied all six items of the mental fatigue subscale of the Work Fatigue Inventory (WFI-3D) by Frone and Tidwell [69]. Instructions, time frame, and response options were the same as for physical fatigue. A sample item is “How often did you feel mentally exhausted at the end of the workday?”

#### 2.3.13. Emotional Fatigue

We applied all six items of the emotional fatigue subscale of the Work Fatigue Inventory (WFI-3D) by Frone and Tidwell [69]. Instructions, time frame, and response options were the same as for physical and mental fatigue. A sample item is “How often did you feel emotionally exhausted at the end of the workday?”

#### 2.3.14. Burnout

We applied all six items of the Personal Burnout subscale of the Copenhagen Burnout Inventory (CBI) by Kristensen et al. [70] to measure burnout. The CBI contains 19 validated items across three subscales, namely Work-Related Burnout, Client-Related Burnout, and Personal Burnout. The CBI has been adapted to for use in German by Nübling et al. [78]. We applied all items of the personal burnout facet. Responses ranged from 1 (*less than once a month*) to 5 (*several times a day*). A sample item is “How often did you feel emotionally exhausted?”

#### 2.3.15. Psychosomatic Complaints

We applied 8 out of the 12 items of the somatic complaints subscale of the Symptom Checklist 90 (SCL-90) by Derogatis [79]. The SCL-90 has been adapted for use in German by Franke [80]. We asked participants to rate to what extent the following health issues had impaired them within the last three months. Responses ranged from 1 (*not at all*) to 5 (*very severely*). Sample items are “headaches”, “faintness or dizziness”, and “pains in the heart or chest”. We confined ourselves to 8 out of the 12 items. Four items overlapped with aspects or concomitants of fatigue, such as “heavy feelings in your arm or legs”.

#### 2.3.16. Satisfaction with Life

We applied all five items of the validated Satisfaction with Life Scale (SWLS) by Diener et al. [81]. The SWLS has been adapted for use in German by Glaesmer et al. [82]. Responses ranged from 1 (*strongly disagree*) to 7 (*strongly agree*). A sample item is “In most ways my life is close to my ideal”.

In Appendix A, we provide a detailed summary of the measures applied.

### 2.4. Analytic Steps for Sample 1

We first examined the bivariate correlations between each of the six OVC items and the composite scores of all aspects of work-related rumination to explore their overlap with one another. In the following, we conducted a set of exploratory factor analyses with item selection. We focused on the aspects of work-related rumination that correlated at 0.70 or higher with at least one of the six OVC items. We set 0.70 as the relevant cut-off value because conventionally correlations of 0.70 or above are considered to be evidence for convergent validity and against discriminant validity [66]. We applied the EFA.dimensions-package version 0.1.7.3 [83] and the psych-package version 2.1.3 [84] in the R statistics environment. In the factor analyses, we applied principal axis factoring (PAF) due to its robustness independent distribution of the underlying variables [85]. Bartlett test (*χ*^2^ = 7098.41, *df* = 276, *p* < 0.01), Kaiser–Meyer–Olkin coefficient (0.96), and the measure of sample adequacy coefficients (ranging from 0.91 to 0.97) indicated that the data were suitable for the analysis. The parallel analysis and the minimum-average-partial test consistently favored a 3-factor model. We drew on a polychoric correlation matrix because estimates tend to be more precise with this approach compared to results drawing on a Pearson correlation matrix [86]. We applied oblimin rotation to allow factors to be correlated. 

## 3. Results

### 3.1. Convergence of OVC with Other Work-Related Rumination Aspects (Sample 1)

In Table 1 we present the descriptive statistics and correlations among all variables considered in Sample 1. We report composite reliabilities (McDonald’s Omega ω) on the diagonal [20]. We have created three composites of OVC, namely the original version consisting of six items and two alternative scores sourced from either the most reliable five or the most reliable four items (according to loadings and communalities). We include the various composite scores for the sake of transparency to allow comparisons to studies applying the six-item version of OVC. In Table 2, we present the correlations between each OVC item and the aspects of work-related rumination introduced above. Items of OVC correlated most strongly with (1) psychological detachment (REQ), (2) distraction (WRRQ), (3) cognitive irritation, and (4) inability to recover (FABA).

### 3.2. Factors Underlying OVC and Selection of Work-Related Rumination Aspects (Sample 1)

We factor-analyzed the 24 items selected from the OVC, psychological detachment, distraction, cognitive irritation, and inability to recover scales. We selected this set of items because at least one of the OVC items correlated above 0.70 with these scales. 

We found that a three-factor solution fitted best. In Table 3 we report the loadings, communalities, and proportion of variance explained of the focal items derived from the three-factor model. The three extracted factors explained 54% of variance. The focal items loaded unambiguously on one of the three factors and did not yield sizeable cross-loadings. The three factors can be interpreted as: (1)inability to unwind from work;(2)psychological detachment;(3)distraction.

The first factor (inability to unwind) was sourced primarily from the six inability to recover items from the FABA, the three cognitive irritation items, and four of the six OVC items. The first OVC item also yielded a small loading on the first factor. The second factor (psychological detachment) was sourced from the four psychological detachment items of the REQ and the third OVC item tapping into ease of switching off after work. The third factor (distraction) was sourced exclusively from the five distraction items from the WRRQ. 

The first OVC item, which explicitly captures time pressure, turned out to be an unreliable indicator of the inability to unwind factor, apparently due to its focus on job stressors rather than problems switching off. The reverse-scored third OVC item capturing ease of switching off turned out to be an indicator of the second factor capturing (ease of) psychological detachment. The sixth OVC item loaded unambiguously, but only moderately, on the first factor capturing inability to unwind. We suggest that the rather modest reliability of this item stems from its focus on unfinished tasks as a cause of sleep impairment—which is a very specific symptom of inability to unwind from work. The first and the sixth OVC items yielded low communalities. Taking both factor loadings and communalities into account, the second, fourth, and fifth OVC items emerged as the most reliable indicators of the factor inability to unwind from work. Although the results of our exploratory factor analysis favor excluding the first and the sixth OVC items, we caution that results of exploratory factor analyses tend to vary considerably from sample to sample. Accordingly, we present different versions of the OVC composite in the correlation tables to report links of more pure or more contaminated variants of OVC and how they relate to outcome variables. 

### 3.3. Implications of the Exploratory Factor Analyses (Sample 1)

Item-level analyses of OVC suggest that at least some of the six OVC items are suboptimal from a psychometric perspective due to too low loadings and cross-loadings. Hence, we recommend removing the first and the sixth item tapping into workload and sleep impairment due to workload to make the scale more internally consistent. The proposed four-item version of the OVC scale avoids confounding distinct aspects in a unidimensional scale. The results reported above suggest that OVC items correlate highly with several aspects of work-related rumination. The core of the scale (three or four out of six items) connotes content overlap with inability to unwind from work in off-job time. Our factor analyses provide evidence that overcommitment essentially captures inability to recover. In the remainder of this paper, we will focus on the four-item version of the OVC scale throughout all analyses. However, we report bivariate correlations for the four-item version, the five-item version, and the six-item version for the sake of transparency.

The results suggest inability to recover, cognitive irritation, and OVC might be applied largely interchangeably to capture inability to unwind from work. Our results support the distinction between inability to unwind and psychological detachment as the potential antipode of inability to unwind. Our finding challenges the practice of treating distraction as an alternative measure and an interchangeable scale of psychological detachment. Besides the slightly different content of the items from these two scales, the different response formats (agreement vs. frequency) may have contributed to the emergence of distinct factors. We revisit the link between psychological detachment and distraction in factor analyses with Sample 2. 

### 3.4. Theoretical Rationale for the Confirmatory Analyses (Sample 2)

Drawing on the results obtained in Sample 1, we set out to locate overcommitment more precisely in the nomological net of work-related rumination constructs. Hence, we applied confirmatory factor analysis to Sample 2 to determine overlap and distinctiveness among overcommitment and the other work-related rumination constructs. That is, we conducted an a priori specification of factors and related item loadings in advance and compared different models with a more nuanced vs. a more parsimonious factor structure. Of note, our analytical facilitates understanding the full range of constructs in this domain and provides a precise description of which measures can be considered more or less unique or redundant in relation to other constructs. Rather than drawing on the specific factor solution found for a subset of 24 items in Sample 1, we adopted a theory-driven approach in Sample 2 by specifying factors consistent with the distinct scales/constructs established and validated in the literature. This approach enabled us to align our findings with the existing literature and provide a theoretical context for our results. However, we also acknowledge the importance of considering the findings from Sample 1. Therefore, we compared models that either combined items of OVC, cognitive irritation, and inability to recover or specified them as distinct factors. This helped us to ensure that our results were consistent with the findings from our initial analysis and provided a comprehensive understanding of the construct of interest. Drawing on this more precise description of the structure underlying the ten work-related rumination constructs, we utilize the data of Sample 2 to examine differential links of each construct to a set of outcomes. 

### 3.5. Factorial Structure Underlying the Ten Work-Related Rumination Constructs (Sample 2)

In Table 4, we present the correlations among the focal variables in Sample 2. We applied confirmatory factor analysis to specify competing factorial structures. We specified a measurement model with all items loading on their respective factors. We tested a 10-factor model consisting of (1) overcommitment, (2) psychological detachment (REQ), (3) affective rumination (WRRQ), (4) problem-solving pondering (WRRQ), (5) positive work reflection, (6) negative work reflection, (7) cognitive irritation, (8) emotional irritation, (9) distraction (WRRQ), and (10) inability to recover (FABA). We freely estimated covariances among all factors across all confirmatory factor analyses. We compared this focal model with a set of competing models. More specifically, we tested a single-factor model with all items loading on a common factor. Furthermore, we specified models combining two or more work-related rumination factors in case these factors were highly correlated. In Appendix A, we illustrate the set of competing factorial structures examined.

We tested these models applying the lavaan package version 0.6–11 [87] for the R statistics environment. We applied the robust maximum likelihood estimator (MLR) to handle non-normally distributed data. We refer to standardized coefficients (loadings, covariances) when reporting results. We report robust estimates of Comparative Fit Index (CFI), Tucker–Lewis Index (TLI), Root Mean Square Error of Approximation (RMSEA), and Standardized Root Mean Square Residual (SRMR). According to Schumacker and Lomax [88], CFI and TLI values above 0.90, RMSEA values below 0.08, and SRMR values below 0.10 signal acceptable fit. We compared nested models by means of the *χ*^2^-difference test to identify the structure that fitted the data best. 

In Table 5, we report the fit indices and comparisons across models. The preferred 10-factor model achieved an acceptable fit to the data (CFI = 0.920, TLI = 0.911, RMSEA = 0.059, and SRMR = 0.058) and fitted the data better than any of the competing models combining either OVC items or cognitive irritation items with the most highly correlated variables. More specifically, combining OVC and psychological detachment in a common factor (Model 9a) did not improve model fit. Combining cognitive irritation and overcommitment (Model 9b), detachment (Model 9e), distraction (Model 9f), and inability to recover (Model 9g) to load on a common factor did not improve model fit. However, the 10-factor model and Model 9b, combining OVC and cognitive irritation, fitted the data equally well. Hence, distinguishing between OVC and cognitive irritation is hardly warranted. 

In Table 6, we report the estimated correlations among the factors from the 10-factor model. It is obvious that several of the ten work-related rumination constructs studied here are very highly correlated. For instance, OVC correlated at Ψ = 0.96 with cognitive irritation, at Ψ = −0.75 with psychological detachment, and at Ψ = 0.74 with inability to recover. Estimated correlations clearly exceeded the threshold of |*r|* < 0.70 for supporting discriminant validity [66]. Although less markedly, the same applies to correlations between OVC and distraction at Ψ = −0.70. 

Psychological detachment correlated at Ψ = 0.92 with cognitive irritation and at Ψ = 0.74 with distraction. The correlation between psychological detachment and inability to recover at Ψ = −0.68 was likewise very high. 

Affective rumination correlated moderately with most aspects of work-related rumination. However, correlations with cognitive irritation and inability to recover were Ψ = 0.67 and Ψ = 0.64, respectively. Problem-solving pondering correlated moderately with most aspects of work-related rumination. The highest correlation emerged as cognitive irritation at Ψ = 0.61.

Positive work reflection was only weakly or not at all related with any of the other work-related rumination constructs with estimated correlations ranging from |Ψ| = 0.07 to |Ψ| = 0.24. Negative work reflection showed weak to moderate links to the other work-related rumination constructs, with correlations ranging from |Ψ| = 0.07 to |Ψ| = 0.41.

Besides the almost perfect correlation between cognitive irritation, OVC, and psychological detachment, estimated correlations of cognitive irritation with distraction and inability to recover ranged from Ψ = 0.86 to Ψ = 0.88. By contrast, emotional irritation yielded weak to moderate links to the other work-related rumination constructs, as reflected in estimated correlations ranging from |Ψ| = 0.21 to |Ψ| = 0.48. 

Inability to recover converged particularly strongly with cognitive irritation and OVC, as reflected in correlations of Ψ = 0.88 and Ψ = 0.74, respectively. The overlap with psychological detachment and affective rumination was also considerable, as reflected in correlations of Ψ = −0.68 and Ψ = 0.64.

To address the issue of whether each of the facets of work-related rumination can be considered unique vs. redundant, we referred to the Fornell–Larcker criterion of convergent vs. discriminant validity [66,89]. Accordingly criterion variables can be considered distinct (indicating discriminant validity) if the squared correlation between two factors or variables is lower than the average variance extracted from the items. In other words, if the items of a given factor explain a larger portion of variance in Factor A than in another Factor B, then there is evidence of discriminant validity. In Table 7, we report the average variance extracted across all facets of work-related rumination and present the squared bivariate correlations. We present squared correlations from observed variables (composite scores) above the diagonal and squared correlations of latent factors as inferred from the confirmatory factor analysis below the diagonal. We found that squared observed correlations are lower than average variance extracted for all facets of work-related rumination. Findings on squared latent correlations draw a very similar picture, but with one exception: Squared correlations of cognitive irritation with (1) OVC, (2) psychological detachment, (3) distraction, and (4) inability to recover are higher than the average variance extracted from these variables. In other words, distinguishing cognitive irritation from these other constructs is probably not warranted. 

In sum, our analysis of the structure underlying the ten work-related rumination constructs suggests that all ten aspects emerge as largely empirically distinct factors. Combining two or three of these constructs with common factors does not improve model fit. Although this finding suggests that distinguishing these ten aspects is warranted from a psychometric perspective, a closer look at the estimated correlations from a pragmatic perspective reveals that some aspects of work-related rumination are barely distinguishable. Some constructs, such as positive and negative work reflection, have negligible overlap with other constructs. The “famous five” facets (detachment, affective rumination, problem-solving pondering, positive work reflection, and negative work reflection) emerge as unambiguously unique factors. However, the less intensively studied measures—namely cognitive irritation, distraction, and inability to recover—essentially measure the same aspects of work-related rumination as does the psychological detachment scale. 

### 3.6. Relative Predictive Power of the Ten Work-Related Rumination Constructs (Sample 2)

The redundancies across the work-related rumination constructs reported above suggest that a more parsimonious structure may suffice to capture the full range of work-related rumination. We set out to compare the predictive validity of the ten work-related rumination constructs vis-à-vis one another. We performed relative weight analyses [90,91,92] to quantify the unique predictive power of each construct regarding six indicators of employee well-being and health, namely (1) physical fatigue, (2) mental fatigue, (3) emotional fatigue, (4) burnout, (5) psychosomatic complaints, and (6) satisfaction with life. Unlike traditional multiple regression models, relative weight analysis explicitly takes into account that predictors may correlate highly among one another [91] and provides the opportunity to partition variance explained across multiple predictors more accurately [93]. In our study, we focused on the relative weight analyses as our primary method but also included traditional regression models as a secondary method of analysis. The regression models were tested independent of the relative weight analyses using R. We examined whether the assumptions and requirements for running regression were met and found no evidence of severe violations. We also include residuals and the Q–Q plots in the Appendix A. Relative weight analysis using the RWAweb application [93] utilizes bootstrapping procedures for the focal tests and hence is not dependent on normally distributed data to be accurate. We report variance explained by each predictor and by the set of predictors as a whole, but our focus was not on finding a set of predictors that optimizes variance explained. Instead, our emphasis was on comparing the relative weight of each facet of work-related rumination vis-à-vis the other facets. Hence, we report the absolute variance explained primarily for descriptive purposes.

In Table 8, Table 9, Table 10, Table 11, Table 12 and Table 13, we report the results separately for each outcome variable. In this section, we describe the results focused on the strongest unique predictors. Table 8 shows that combining the ten work-related rumination constructs as predictors explained about thirty percent of variance in physical fatigue. A closer look at the relative weights (RW) and the rescaled relative weights (RS-RW%) suggests that each facet explained some variance in physical fatigue. However, only a few constructs explained large portions of the variance explainable by all constructs. Emotional irritation, affective rumination, and negative work reflection each explained approximately six percent of the variance in physical fatigue. Positive work reflection and inability to recover each explained about three percent of the variance in physical fatigue. The other five aspects of work-related rumination explain small portions of unique variance. For instance, psychological detachment uniquely explains only about 1.5 percent of variance in physical fatigue. 

A similar pattern of results was evident in Table 9, focusing on mental fatigue. In total, the ten work-related rumination constructs explained thirty-six percent of variance in mental fatigue. Again, emotional irritation, affective rumination, inability to recover, and negative work reflection were among the strongest predictors. Emotional irritation alone explained approximately ten percent of variance (a fourth of the variance explained by all predictors). Affective rumination, inability to recover, negative work reflection, and cognitive irritation each explained approximately five percent of variance. 

Considering emotional fatigue as outcome (Table 10), the ten work-related rumination constructs explain thirty-one percent of variance. Emotional irritation, affective rumination, inability to recover, and negative work reflection are the strongest predictors. Emotional irritation alone explains approximately eight percent of variance (a fourth of the variance explained by all predictors). Inability to recover, affective rumination, and negative work reflection explain, respectively, five, four, and three percent of unique variance. Positive work reflection and cognitive irritation explain roughly two percent of variance in emotional fatigue. 

As shown in Table 11, the ten work-related rumination constructs explained forty-five percent of variance in burnout. Like the results for emotional fatigue, emotional irritation, affective rumination, inability to recover, and negative work reflection were among the strongest predictors. Emotional irritation alone explained approximately twelve percent of variance (a quarter of the variance explained by all predictors). Affective rumination uniquely explained seven percent of variance in burnout. Inability to recover, negative work reflection, and positive work reflection each explained five percent of unique variance. Psychological detachment and distraction each explained two percent of unique variance in burnout. 

Table 12 presents the results for psychosomatic complaints, for which, the ten work-related rumination constructs explained twenty-four percent of the variance. Emotional irritation, affective rumination, and inability to recover were the strongest predictors. Emotional irritation alone explained approximately seven percent of the variance (a fourth of the variance explained by all predictors). Affective rumination uniquely explained six percent of the variance in psychosomatic complaints. Inability to recover and overcommitment explained, respectively, four and two percent of the unique variance in psychosomatic complaints. Negative work reflection, cognitive irritation, and distraction explained less than two percent of the unique variance in psychosomatic complaints.

In Table 13, we present the results of the relative weight analysis predicting satisfaction with life with eighteen percent of explained variance. Positive work reflection and emotional irritation emerged as the strongest predictors. Positive work reflection alone explained approximately six percent of variance (a third of the variance explained by all predictors). Emotional irritation explained five percent of unique variance in satisfaction with life. Overcommitment, affective rumination, and inability to recover explained approximately one percent of unique variance in satisfaction with life. 

Figure 1 shows a graphic summary of the RWA across outcomes. These results provide a precise description of the relative relevance of each work-related rumination construct studied here and helps to prioritize and select constructs from a pragmatic perspective with predictive validity as the core criterion. Although the specific results across the six outcomes varied considerably, several findings replicate consistently. First, emotional irritation is consistently among the strongest unique predictors across outcomes explaining one fourth to one third of the variance explainable by the ten work-related rumination constructs. Although the effects were less pronounced and less consistent, the same applies to affective rumination and inability to recover. Both constructs emerged as the second or third best predictor of the six outcomes. Negative work reflection and positive work reflection tended to contribute uniquely to the prediction of outcomes, supporting these constructs as valuable aspects within the “famous five” facets of work-related rumination. Of note, when it comes to predicting positively connoted aspects of employee well-being, positive work reflection plays a vital role. OVC explained less than two percent of unique variance and hence added little to the prediction of the six outcomes of interest when considering other constructs tapping into inability to unwind from work. A similar pattern emerged for psychological detachment and distraction. This finding is noteworthy given that psychological detachment has been studied more extensively than any of the other constructs. We suggest that the common theme of inability to unwind from work (as captured in psychological detachment, cognitive irritation, distraction, and inability to recover) is relevant for predicting fatigue, burnout, psychosomatic complaints, and satisfaction with life. 

## 4. Discussion

In our study, we examined the nomological network of work-related rumination. Across two samples, we ran factor analyses for OVC items along with items from nine work-related rumination constructs. We compared criterion-related validity regarding well-being and health outcomes across the ten constructs, applying multiple regression analyses and relative weight analyses. 

### 4.1. Theoretical Implications

Our study provides empirical evidence on the position of OVC within the nomological network of work-related rumination constructs. OVC overlaps highly with psychological detachment, cognitive irritation, distraction, and inability to recover as reflected in correlations above 0.80 or even 0.90, which suggests that these measures can be used largely interchangeably.

Our results confirm the distinction of the “famous five” facets of work-related rumination (psychological detachment, affective rumination, problem-solving pondering, positive work reflection, and negative work reflection) as unique and unambiguously non-redundant constructs with moderate intercorrelations. Our findings are consistent with recent integrative research [9]. However, several less extensively studied work-related rumination constructs converge strongly with psychological detachment, especially when considering latent factors. Hence, our analyses suggest that it may not make a big difference whether ease of unwinding or inability to unwind is focal in the questionnaire items when capturing aspects of work-related rumination. 

In their recent meta-analysis on work-related rumination facets, Jimenez et al. [8] have proposed a continuum of work-related rumination constructs ranging from pure to contaminated constructs Our study reveals that contaminated constructs, such as affective rumination or positive work reflection, contribute uniquely to the prediction of well-being outcomes. These findings are consistent with conceptual arguments and meta-analytic evidence [8] that being “switched on” during off-job time per se may not be detrimental to well-being and health [1]. 

These results are consistent with existing evidence on the unique benefits of problem-solving pondering and positive work reflection for satisfaction with life, flourishing, and thriving [9]. 

Our relative weight analyses suggest that emotional irritation and affective rumination are the most important predictors consistently across the outcomes considered. This may imply that focusing on these aspects of work-related rumination is warranted when trying to optimize prediction of employee well-being and health. At this point, we would like to caution against this shortcut. Irritation has been conceptualized explicitly as an indicator of strain or impaired well-being rather than a measure of work-related rumination. However, the strong correlation between emotional irritation and affective rumination made us consider emotional irritation as an aspect of work-related rumination. Weigelt et al. [9] reported that emotional irritation (but not affective rumination) correlates strongly with neuroticism and may, therefore, be conceptually less precise and more ambiguous than desired. From a conceptual point of view, the “famous five” facets are probably less ambiguous and clearer than the less extensively studied additional facets (cognitive irritation, emotional irritation, distraction, and inability to recover), as reflected in the moderate intercorrelations and differential links with outcomes [9]. 

### 4.2. Practical Implications

Our study provides evidence-based guidance on navigating through the landscape of work-related rumination constructs and measures. Our results aid researchers in selecting scales for research on work-related rumination, its antecedents, its outcomes, and contingencies underlying health-impairing effects of work as conceptualized in the perseverative cognition hypothesis [4]. Researchers can consult the correlation tables provided here to select scales with minimal redundancy. 

We have argued above that it may not matter whether a scale captures ease of psychological detachment or inability to unwind from a psychometric perspective. However, measures referring explicitly to ruminative thoughts or inability to unwind may prompt rumination. Hence, although measures such as inability to recover may optimize predictive validity, focusing on psychological detachment rather than inability to recover may be a viable option. 

### 4.3. Strengths and Limitations

Our research should be interpreted in the light of a few limitations. We have drawn on cross-sectional survey data. Accordingly, our research applies only to the between-person level (i.e., habitual and stable differences between persons). This perspective is consistent with research on OVC. However, a large volume of research on work-related rumination has applied experience sampling methodology and focused on effects at the within-person level. We cannot claim that our findings are generalizable to the within-person level. However, a cross-sectional survey study is a reasonable first step that allows us to draw conclusions regarding the original persistent or trait-framed constructs. Moreover, our focus was on assessing convergence vs. uniqueness across the ten work-related rumination constructs and comparing the relative predictive validity across constructs (rather than establishing it for a single predictor). Cross-sectional survey studies are eminently suitable for these aims [94], and our approach provided the opportunity to include and concurrently study a large set of constructs and comprehensive sets of items. 

We did not consider several aspects of work-related rumination, such as basking (i.e., positive affect while thinking about work) [95] and wallowing (i.e., looking back at successes at work) [96]. However, positive work reflection may overlap considerably with these constructs as reflected in correlations between positive work reflection and positive affect [97]. 

We relied exclusively on self-reports because the focal work-related rumination measures are self-report scales. In this case, method bias may inflate correlations among variables [98,99,100]. However, we suggest that inflated correlations among variables may render our examination of the structure underlying work-related rumination constructs more conservative as it may be harder to find evidence that the factors are indeed distinct. 

In the regression models and the relative weight analyses, we found that the ten facets of work-related rumination, when considered together, explain a limited portion of variance in the criterion variables, ranging from 17 to 45 percent. This finding highlights that there are several other variables beyond work-related rumination that also contribute to this variance. Hence, it is crucial to consider the relative importance of work-related rumination in comparison to other meaningful predictors such as job characteristics or lifestyle risk factors in future studies.

### 4.4. Avenues for Future Research

We strongly encourage research that studies the role of psychological detachment in concert with the other aspects of the “famous five” to provide insights into the unique relevance of each aspect regarding specific outcomes. We have taken a first step in this direction, but experience sampling research on work-related rumination has just begun to explore more than one aspect of rumination as it relates to next-day well-being and performance [101].

Research explicit on time perspective and valence regarding work-related thoughts is still in a nascent state [102]. Rutten and colleagues [103] recently developed an instrument tailored to capture anticipatory work-related thoughts (work prospection). In a similar vein, Noja et al. [104] proposed a measure of work–home integration tapping into past- vs. future-oriented thoughts about work during off-job time. 

Further exploration of the structure of work-related rumination might be worthwhile, for instance, relating the emergence of a common factor underlying the scales range (R-factor [105]), higher-order valence (positive vs. negative thought content), or mode factors (emphasis on affect or cognition [106]). In recent years, researchers have utilized bi-factor models to capture a general factor of constructs such as basic need satisfaction [107]. 

Facets of work-related rumination are commonly included in large-scale panel studies representative of the population, such as the Socio-Economic Panel in Germany [108], which affords an opportunity to run full structural equation models, including measurement and structural models in the future [109].

## 5. Conclusions

We examined ten facets of work-related rumination during off-job time with an emphasis on OVC as a construct that is closely related to aspects of work-related rumination, but that has rarely been empirically linked to this research stream. Drawing on self-report survey data, we performed factor analysis to determine the position of OVC in the nomological network of work-related rumination constructs. Our analyses suggest that although OVC is technically distinct from the other facets of work-related rumination, the overlap with psychological detachment is very high. Hence, distinguishing OVC from psychological detachment may not add much value from a practical perspective. We suggest treating OVC, psychological detachment, cognitive irritation, distraction, and inability to recover as alternative reflections of the same underlying construct. Our study replicates evidence that favors basically five unambiguously distinct facets of work-related rumination, namely (1) psychological detachment, (2) affective rumination, (3) problem-solving pondering, (4) positive work reflection, and (5) negative work reflection. Applying relative weight analyses to compare the unique predictive power of each work-related rumination facet regarding major well-being and health outcomes, we found that facets that explicitly tap into the content and valence of thoughts, and the affective experiences accompanying these thoughts, such as affective rumination, emotional irritation, and positive work reflection, explained the largest proportions of variance in well-being and health. Our study draws a precise and nuanced picture of the work-related rumination construct landscape and especially the connections among constructs and the unique features of specific facets. Our research paves the way for more integrated and coherent research on the full range of work-related rumination from OVC to psychological detachment. 

## Figures and Tables

**Figure 1 ijerph-20-03573-f001:**
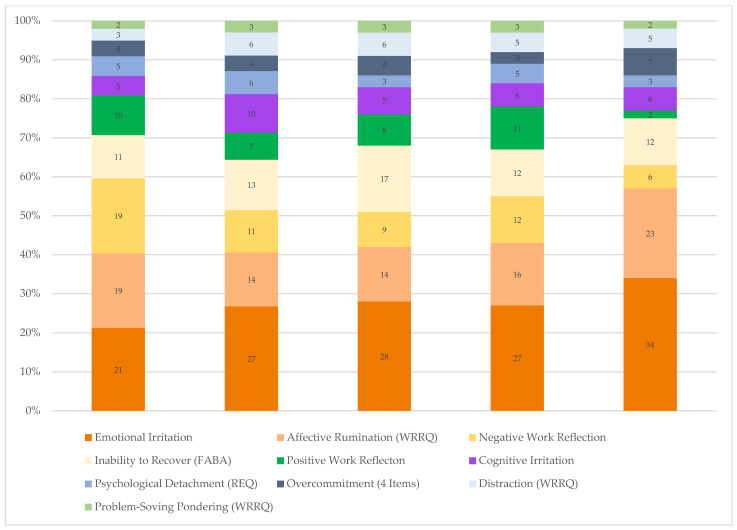
Work-Related Rumination Constructs and Relative Explained Variance in Well-being and Health Outcomes. Note. Overcommitment (4 Items) = composite of items 2, 3, 4, and 5; REQ = recovery experience questionnaire; WRRQ = work-related rumination questionnaire; FABA = faulty attitudes and behaviors analysis.

**Table 1 ijerph-20-03573-t001:** Means, standard deviations, and zero-order correlations among study variables (Sample 1).

	Variable	*M*	*SD*	1	2	3	4	5	6	7	8	9	10	11	12	13	14	15	16
1	Age	32.31	9.64																
2	Gender	-	-	−0.02															
3	Working hours per week	27.89	10.80	0.22 ***	−0.02														
4	Leadership position	-	-	0.15 **	−0.09	0.22 ***													
5	Overcommitment (4 Items)	2.68	0.94	0.16 **	0.16 **	0.23 ***	0.12 *	(0.85)											
6	Overcommitment (5 Items)	2.71	0.87	0.13 *	0.18 **	0.22 ***	0.11 *	0.97 ***	(0.82)										
7	Overcommitment (6 Items)	2.73	0.80	0.14 *	0.20 ***	0.21 ***	0.12 *	0.95 ***	0.98 ***	(0.80)									
8	Psychological Detachment (REQ)	3.28	1.01	−0.09	−0.08	−0.24 ***	−0.14 **	−0.75 ***	−0.72 ***	−0.69 ***	(0.92)								
9	Affective Rumination (WRRQ)	2.47	0.94	−0.02	0.13 *	0.11 *	0.08	0.56 ***	0.58 ***	0.59 ***	−0.48 ***	(0.90)							
10	Problem-Solving Pondering (WRRQ)	2.64	0.84	0.18 **	0.04	0.14 **	0.12 *	0.63 ***	0.63 ***	0.61 ***	−0.59 ***	0.41 ***	(0.83)						
11	Positive Work Reflection	2.70	0.91	0.06	−0.08	−0.09	0.04	0.02	0.02	0.01	−0.08	−0.22 ***	0.26 ***	(0.91)					
12	Negative Work Reflection	2.79	0.94	−0.07	0.07	0.08	−0.03	0.42 ***	0.41 ***	0.42 ***	−0.36 ***	0.52 ***	0.35 ***	−0.01	(0.92)				
13	Cognitive Irritation	2.57	1.03	0.11 *	0.11 *	0.23 ***	0.15 **	0.86 ***	0.85 ***	0.83 ***	−0.79 ***	0.63 ***	0.64 ***	0.00	0.47 ***	(0.91)			
14	Emotional Irritation	2.31	0.93	−0.03	0.10	0.09	0.01	0.41 ***	0.43 ***	0.44 ***	−0.30 ***	0.62 ***	0.23 ***	−0.23 ***	0.38 ***	0.46 ***	(0.91)		
15	Distraction (WRRQ)	3.26	1.00	−0.07	−0.14 **	−0.17 **	−0.04	−0.72 ***	−0.72 ***	−0.71 ***	0.76 ***	−0.55 ***	−0.56 ***	0.02	−0.40 ***	−0.76 ***	−0.36 ***	(0.91)	
16	Inability to Recover (FABA)	2.31	0.97	0.09	0.10	0.19 ***	0.14 **	0.76 ***	0.77 ***	0.78 ***	−0.65 ***	0.65 ***	0.61 ***	−0.07	0.53 ***	0.80 ***	0.47 ***	−0.72 ***	(0.87)

Note. *n* = 357. *M* = mean; *SD* = standard deviation; REQ = recovery experience questionnaire; WRRQ = work-related rumination questionnaire; FABA = faulty attitude and behavior analysis. McDonald’s Omega (ω) is on the diagonal in parentheses. Overcommitment (4 Items) = composite of items 2, 3, 4, and 5. Overcommitment (5 Items) = composite of items 2, 3, 4, 5, and 6. Overcommitment (6 Items) = composite of items 1, 2, 3, 4, 5, and 6. We provide confidence intervals for all correlations in Table S4 in the Supplementary Materials. * *p* < 0.05. ** *p* < 0.01. *** *p* < 0.001.

**Table 2 ijerph-20-03573-t002:** Correlations of each OVC item with the Scores of the other Variables (Sample 1).

	Variable	*M*	*SD*	1	2	3	4	5	6
1	OVC Item 1	2.83	1.02						
2	OVC Item 2	2.75	1.14	0.39 **					
3	OVC Item 3 (inverted)	2.77	1.16	0.23 **	0.54 **				
4	OVC Item 4	2.68	1.18	0.27 **	0.40 **	0.36 **			
5	OVC Item 5	2.57	1.15	0.28 **	0.76 **	0.67 **	0.53 **		
6	OVC Item 6	2.81	1.22	0.25 **	0.37 **	0.26 **	0.21 **	0.39 **	
7	Psychological Detachment (REQ)	3.28	1.00	−0.21 **	−0.61 **	−0.75 **	−0.38 **	−0.71 **	−0.26 **
8	Affective Rumination (WRRQ)	2.48	0.94	0.32 **	0.55 **	0.41 **	0.31 **	0.56 **	0.35 **
9	Problem-Solving Pondering (WRRQ)	2.64	0.85	0.18 **	0.56 **	0.48 **	0.43 **	0.59 **	0.33 **
10	Positive Work Reflection	2.70	0.91	−0.06	−0.07	0.04	0.03	0.01	0.02
11	Negative Work Reflection	2.79	0.94	0.23 **	0.46 **	0.35 **	0.23 **	0.37 **	0.17 **
12	Cognitive Irritation	2.56	1.03	0.31 **	0.74 **	0.72 **	0.48 **	0.85 **	0.41 **
13	Emotional Irritation	2.30	0.93	0.27 **	0.35 **	0.33 **	0.26 **	0.38 **	0.28 **
14	Distraction (WRRQ)	3.26	1.00	−0.28 **	−0.62 **	−0.66 **	−0.38 **	−0.71 **	−0.33 **
15	Inability to Recover (FABA)	2.30	0.97	0.39 **	0.66 **	0.62 **	0.50 **	0.70 **	0.41 **

Note. *n* = 357. M = mean; SD = standard deviation; REQ = recovery experience questionnaire; WRRQ = work-related rumination questionnaire; FABA = faulty attitudes and behaviors analysis. ** *p* < 0.01.

**Table 3 ijerph-20-03573-t003:** Results of the Exploratory Factor Analyses in Sample 1.

Origin		Item	*PA1*	*PA2*	*PA3*	*h* ^2^
Overcommitment	1	I get easily overwhelmed by time pressures at work.	**0.47**	0.17	−0.11	0.20
	2	As soon as I get up in the morning, I start thinking about work problems.	**0.61**	−0.26	0.00	0.63
	3	*When I get home, I can easily relax and ‘switch off’ work.*	0.25	**−0.53**	−0.13	0.67
	4	People close to me say I sacrifice too much for my job.	**0.58**	0.00	0.11	0.33
	5	Work rarely lets me go; it is still on my mind when I go to bed.	**0.55**	−0.37	0.00	0.79
	6	If I postpone something that I was supposed to do today I will have trouble sleeping at night.	**0.57**	0.12	0.00	0.29
Psychological Detachment (REQ)	1	During time after work, I forget about work.	0.00	**0.82**	0.14	0.82
	2	During time after work, I do not think about work at all.	0.00	**0.83**	0.00	0.78
	3	During time after work, I distance myself from my work.	−0.13	**0.71**	0.17	0.86
	4	During time after work, I get a break from the demands of work.	0.00	**0.66**	0.24	0.70
Distraction (WRRQ)	1	I am able to stop thinking about work-related matters in my free time.	0.00	0.00	**0.75**	0.63
	2	I find it easy to unwind after work.	−0.18	0.00	**0.72**	0.66
	3	As soon as I leave work, I make myself switch off from work.	0.00	0.00	**0.94**	0.87
	4	I leave work issues behind when I leave work.	0.00	0.00	**0.86**	0.88
	5	I do not feel able to switch off from work.	**−0.52**	0.17	0.32	0.82
Cognitive Irritation	1	I have difficulties relaxing after work.	**0.58**	−0.35	−0.11	0.88
	2	Even at home I often think of my problems at work.	**0.60**	−0.40	0.00	0.80
	3	Even on my vacations I think about my problems at work.	0.49	−0.46	0.00	0.76
Inability to Recover (FABA)	1	Hard to relax because of work	**0.81**	0.12	−0.13	0.69
	2	Sleep impaired due to work-related rumination	**0.88**	0.17	−0.12	0.74
	3	Hard to find time for personal issues	**0.66**	0.00	0.00	0.38
	4	Thinking about work during vacations	**0.64**	−0.20	−0.13	0.78
	5	Striving harder than feasible on the long run	**0.80**	0.00	0.00	0.56
	6	Hard to switch off	**0.65**	−0.22	−0.18	0.90
Percentage of variance			26	15	13	

Note. Items presented in italics entered the analysis coded invertedly. Highest loading per item is set in bold. PA = principal axis. h^2^ = communality coefficient. REQ = recovery experience questionnaire; WRRQ = work-related rumination questionnaire; FABA = faulty attitudes and behaviors analysis.

**Table 4 ijerph-20-03573-t004:** Means, standard deviations, and zero-order correlations among study variables.

	Variable	*M*	*SD*	1	2	3	4	5	6	7	8	9	10	11	12	13	14	15	16	17	18	19	20	21	22
1	Age	32.10	9.53																						
2	Gender	-	-	−0.10																					
3	Working hours per week	27.20	12.20	0.37 ***	−0.14																				
4	Leadership position	-	-	0.35 ***	−0.12 **	0.18 ***																			
5	Overcommitment (4 Items)	2.71	0.96	0.11 *	0.17 ***	0.18 ***	0.21 ***	(0.83)																	
6	Overcommitment (5 Items)	2.71	0.90	0.07	0.20 ***	0.15 **	0.19 ***	0.97 ***	(0.81)																
7	Overcommitment (6 Items)	2.73	0.85	0.06	0.21 ***	0.14 **	0.19 ***	0.96 ***	0.98 ***	(0.82)															
8	Psychological Detachment (REQ)	3.26	1.02	−0.06	−0.07	−0.14 **	−0.19 ***	−0.73 ***	−0.70 ***	−0.70 ***	(0.91)														
9	Affective Rumination (WRRQ)	2.52	0.97	−0.02	0.17 ***	0.05	0.07	0.62 ***	0.64 ***	0.65 ***	−0.47 ***	(0.91)													
10	Problem-Solving Pondering (WRRQ)	2.64	0.87	0.16 **	0.10	0.10 *	0.23 ***	0.58 ***	0.58 ***	0.58 ***	−0.50 ***	0.40 ***	(0.83)												
11	Positive Work Reflection	2.68	0.97	0.01	0.03	−0.12 *	−0.03	−0.11 *	−0.12 *	−0.13 *	0.10 *	−0.29 ***	0.16 **	(0.93)											
12	Negative Work Reflection	2.92	0.98	−0.08	0.13 *	0.07	0.02	0.40 ***	0.40 ***	0.41 ***	−0.32 ***	0.45 ***	0.30 ***	−0.09	(0.92)										
13	Cognitive Irritation	2.58	1.09	0.07	0.12 *	0.17 **	0.19 ***	0.85 ***	0.85 ***	0.84 ***	−0.79 ***	0.62 ***	0.62 ***	−0.11 *	0.42 ***	(0.90)									
14	Emotional Irritation	2.32	1.00	−0.11 *	0.09	0.01	0.02	0.41 ***	0.44 ***	0.46 ***	−0.36 ***	0.56 ***	0.25 ***	−0.27 ***	0.35 ***	0.47 ***	(0.91)								
15	Distraction (WRRQ)	3.25	1.04	−0.08	−0.11 *	−0.08	−0.19 ***	−0.74 ***	−0.72 ***	−0.71 ***	0.73 ***	−0.56 ***	−0.53 ***	0.09	−0.39 ***	−0.79 ***	−0.38 ***	(0.90)							
16	Inability to Recover (FABA)	2.38	0.96	0.06	0.21 ***	0.15 **	0.22 ***	0.81 ***	0.82 ***	0.84 **	−0.68 ***	0.71 ***	0.58 ***	−0.17 **	0.49 ***	0.82 ***	0.52 ***	−0.74 ***	(0.87)						
17	Physical Fatigue (WFI)	3.20	0.90	−0.04	0.28 ***	0.04	−0.02	0.32 ***	0.32 ***	0.36 ***	−0.29 ***	0.46 ***	0.15 **	−0.26 ***	0.37 ***	0.33 ***	0.42 ***	−0.28 ***	0.43 ***	(0.94)					
18	Mental Fatigue (WFI)	3.25	0.94	−0.11 *	0.23 ***	0.02	0.00	0.41 ***	0.42 ***	0.45 ***	−0.35 ***	0.48 ***	0.20 ***	−0.24 ***	0.37 ***	0.45 ***	0.49 ***	−0.39 ***	0.51 ***	0.63 ***	(0.95)				
19	Emotional Fatigue (WFI)	2.80	1.08	0.00	0.23 ***	0.08	−0.02	0.35 ***	0.36 ***	0.38 ***	−0.27 ***	0.45 ***	0.19 ***	−0.23 ***	0.31 ***	0.38 ***	0.45 ***	−0.33 ***	0.49 ***	0.50 ***	0.71 ***	(0.96)			
20	Personal Burnout (CBI)	2.77	1.06	−0.09	0.19 ***	0.01	−0.04	0.41 ***	0.42 ***	0.45 ***	−0.37 ***	0.55 ***	0.23 ***	−0.32 ***	0.41 ***	0.44 ***	0.55 ***	−0.40 ***	0.53 ***	0.67 ***	0.68 ***	0.61 ***	(0.93)		
21	Psychosomatic Complaints (SCL-90)	1.89	0.67	−0.16 **	0.24 ***	−0.08	−0.07	0.34 ***	0.35 ***	0.36 ***	−0.24 ***	0.43 ***	0.13 **	−0.14 **	0.26 ***	0.32 ***	0.41 ***	−0.28 ***	0.41 ***	0.53 ***	0.47 ***	0.45 ***	0.55 ***	(0.80)	
22	Satisfaction with Life (SWLS)	4.60	1.24	0.01	0.02	0.02	0.10	−0.02	−0.02	−0.03	0.09	−0.18 ***	0.04	0.29 ***	−0.12 *	−0.08	−0.29 ***	0.14 **	−0.16 **	−0.24 ***	−0.28 ***	−0.28 ***	−0.37 ***	−0.19 ***	(0.88)

Note. *n* = 388. *M* = mean; *SD* = standard deviation; REQ = recovery experience questionnaire; WRRQ = work-related rumination questionnaire; FABA = faulty attitude and behavior analysis; WFI = 3D-work fatigue inventory; CBI = Copenhagen burnout inventory; SCL-90 = symptom checklist 90; SWLS = satisfaction with life scale. McDonald’s Omega (ω) reliabilities are on the diagonal in parentheses. Overcommitment (4 Items) = composite of items 2, 3, 4, and 5. Overcommitment (5 Items) = composite of items 2, 3, 4, 5, and 6. Overcommitment (6 Items) = composite of items 1, 2, 3, 4, 5, and 6. We provide confidence intervals for all correlations in Table S5 in the Supplementary Materials. * *p* < 0.05. ** *p* < 0.01. *** *p* < 0.001.

**Table 5 ijerph-20-03573-t005:** Fit Indices of the Multi-level Confirmatory Factor Analysis Models in Sample 2.

	*χ* ^2^	*df*	CFI	TLI	RMSEA	SRMR	Δ*χ*^2^	Δ*df*	*sig.*
10-factor model	1810.039	815	0.920	0.911	0.059	0.058			
9-factor model a(OVC + DET)	1993.529	824	0.905	0.896	0.063	0.061	148.730	9	***
9-factor model b(OVC + CIR)	1821.130	824	0.919	0.912	0.059	0.058	11.938	9	
9-factor model c(OVC + DIS)	1939.316	824	0.910	0.901	0.062	0.058	99.615	9	***
9-factor model d(OVC + IAR)	1858.037	824	0.916	0.908	0.060	0.059	42.727	9	***
9-factor model e(DET + CIR)	2005.771	824	0.904	0.895	0.064	0.060	155.510	9	***
9-factor model f(CIR + DIS)	1950.915	824	0.909	0.900	0.062	0.058	112.740	9	***
9-factor model gCIR + IAR)	1889.501	824	0.914	0.906	0.061	0.060	71.526	9	***
8-factor model a(OVC + DET + CIR)	2036.493	832	0.902	0.894	0.064	0.061	197.010	17	***
8-factor model b(OVC + DIS + CIR)	1961.296	832	0.908	0.901	0.062	0.058	130.460	17	***
7-factor model(OVC + DET + CIR + IAR)	2174.829	839	0.892	0.883	0.067	0.061	309.390	24	***
6-factor model(OVC + DET + CIR + DIR + IAR)	2295.495	845	0.882	0.874	0.070	0.064	412.530	30	***
Single-factor model	5606.921	860	0.606	0.586	0.127	0.110	2526.500	45	***

Note. OVC = overcommitment. DET = detachment. CIR = cognitive irritation. DIS = distraction. IAR = inability to recover. *χ*^2^ represents the results of the Chi^2^ test (scaled). *df* represents the degrees of freedom (scaled). CFI represents the Comparative Fit index (robust). TLI represents the Tucker–Lewis Index (robust). RMSEA represents the Root Mean Square Error of Approximation (robust). SRMR represents the Standardized Root Mean Square Residual. Δ*χ*^2^ and Δ*df* represent respective changes in Chi^2^ test results and degrees of freedom relative to the 10-factor model. *sig.* represents significance level of *p*-values. *** *p* < 0.001. The letters a to g indicate that models refer to different structures with an identical number of factors in total.

**Table 6 ijerph-20-03573-t006:** Estimated Covariances among Factor in the Confirmatory Factor Analysis (Sample 2).

	Variable	1	2	3	4	5	6	7	8	9	10
1	Overcommitment (4 Items)										
2	Psychological Detachment (REQ)	−0.75									
3	Affective Rumination (WRRQ)	0.58	−0.47								
4	Problem-Solving Pondering (WRRQ)	0.51	−0.45	0.38							
5	Positive Work Reflection	−0.11	0.10	−0.24	0.05						
6	Negative Work Reflection	0.39	−0.33	0.41	0.28	−0.07					
7	Cognitive Irritation	0.96	−0.92	0.67	0.61	−0.11	0.46				
8	Emotional Irritation	0.35	−0.33	0.48	0.21	−0.21	0.30	0.45			
9	Distraction (WRRQ)	−0.70	0.74	−0.51	−0.47	0.07	−0.37	−0.86	−0.32		
10	Inability to Recover (FABA)	0.74	−0.68	0.64	0.49	−0.13	0.42	0.88	0.40	−0.70	

Note. *n* = 388. REQ = recovery experience questionnaire; WRRQ = work-related rumination questionnaire; FABA = faulty attitudes and behaviors analysis. Overcommitment (4 Items) = composite of items 2, 3, 4, and 5. We also report the factor loadings across all items in Appendix A.

**Table 7 ijerph-20-03573-t007:** Average variance extracted and squared correlations among variables in Sample 2.

	Variable	AVE	1	2	3	4	5	6	7	8	9	10
1	Overcommitment (4 Items)	0.53		0.54	0.38	0.33	0.01	0.16	0.72	0.17	0.54	0.66
2	Psychological Detachment (REQ)	0.71	0.57		0.22	0.25	0.01	0.10	0.63	0.13	0.54	0.46
3	Affective Rumination (WRRS)	0.67	0.34	0.22		0.16	0.08	0.20	0.34	0.32	0.31	0.51
4	Problem-Solving Pondering (WRRS)	0.55	0.26	0.20	0.15		0.03	0.09	0.39	0.06	0.28	0.33
5	Positive Work Reflection	0.76	0.01	0.01	0.06	0.00		0.01	0.01	0.07	0.01	0.03
6	Negative Work Reflection	0.74	0.15	0.11	0.17	0.08	0.00		0.18	0.12	0.15	0.24
7	Cognitive Irritation	0.75	0.91	0.85	0.44	0.38	0.01	0.21		0.22	0.63	0.68
8	Emotional Irritation	0.66	0.12	0.11	0.23	0.04	0.04	0.09	0.21		0.14	0.27
9	Distraction (WRRS)	0.65	0.49	0.55	0.26	0.22	0.01	0.14	0.73	0.10		0.54
10	Inability to Recover (FABA)	0.59	0.55	0.46	0.41	0.24	0.02	0.18	0.77	0.16	0.49	

Note. *n* = 388. AVE = average variance extracted. Overcommitment (4 Items) = composite of items 2, 3, 4, and 5; REQ = recovery experience questionnaire; WRRQ = work-related rumination questionnaire; FABA = faulty attitude and behavior analysis. Below the diagonal, we report the squared correlations of latent covariances as estimated in the confirmatory factor analysis. Above the diagonal we report the squared correlations of observed variables (unweighted composite scores).

**Table 8 ijerph-20-03573-t008:** Coefficients of the relative weight analyses predicting physical fatigue (Sample 2).

Criterion	Physical Fatigue									
Predictor	*b*	*SE*	*T*	*p*		RW		CI-L	CI-U	RS-RW (%)
Intercept	2.17	0.47	4.59	0.00	***					
Overcommitment (4 Items)	−0.03	0.08	−0.38	0.70		0.0163		0.0083	0.0265	5.32
Psychological Detachment (REQ)	−0.10	0.07	−1.47	0.14		0.0155		0.0056	0.0314	5.05
Affective Rumination (WRRQ)	0.18	0.06	2.79	0.01	**	0.0594	*	0.0304	0.0971	19.34
Problem-Solving Pondering (WRRQ)	−0.09	0.06	−1.44	0.15		0.0055		0.0021	0.0068	1.81
Positive Work Reflection	−0.09	0.04	−2.08	0.04	*	0.0290	*	0.0068	0.0632	9.45
Negative Work Reflection	0.17	0.05	3.59	0.00	***	0.0547	*	0.0237	0.0983	17.82
Cognitive Irritation	−0.05	0.09	−0.60	0.55		0.0160	*	0.0086	0.0232	5.20
Emotional Irritation	0.15	0.05	3.09	0.00	**	0.0602	*	0.029	0.0994	19.61
Distraction (WRRQ)	0.08	0.07	1.28	0.20		0.0107		0.0056	0.0158	3.48
Inability to Recover (FABA)	0.21	0.09	2.38	0.02	*	0.0396	*	0.0212	0.0615	12.90
Adjusted *R*-squared						0.2882				
*F* [10, 377]						16.47				
*p*						<0.001				

Note. *n* = 388. b = unstandardized regression coefficient; SE = standard error; RW = raw relative weight (within rounding error, raw weights will sum to R-squared); CI-L = lower bound of confidence interval used to test the statistical significance of raw weight; CI-U = upper bound of confidence interval used to test the statistical significance of raw weight; RS-RW (%) = relative weight rescaled as a percentage of predicted variance in the criterion variable attributed to each predictor (within rounding error, rescaled weights sum to 100%). Overcommitment (4 Items) = composite of items 2, 3, 4, and 5; REQ = recovery experience questionnaire; WRRQ = work-related rumination questionnaire; FABA = faulty attitude and behavior analysis. * *p* < 0.05. ** *p* < 0.01. *** *p* < 0.001.

**Table 9 ijerph-20-03573-t009:** Coefficients of the relative weight analyses predicting mental fatigue (Sample 2).

Criterion	Mental Fatigue									
Predictor	*b*	*SE*	*T*	*p*		RW		CI-L	CI-U	RS-RW (%)
Intercept	2.00	0.48	4.15	0.00	***					
Overcommitment (4 Items)	−0.03	0.09	−0.38	0.70		0.0267		0.0153	0.0401	7.38
Psychological Detachment (REQ)	0.01	0.07	0.09	0.93		0.0191		0.0100	0.0322	5.27
Affective Rumination (WRRQ)	0.08	0.06	1.25	0.21		0.0483	*	0.0257	0.0751	13.34
Problem-Solving Pondering (WRRQ)	−0.14	0.06	−2.22	0.03	*	0.0082		0.0043	0.0099	2.26
Positive Work Reflection	−0.07	0.05	−1.54	0.13		0.0211		0.004	0.0496	5.84
Negative Work Reflection	0.12	0.05	2.50	0.01	*	0.0407	*	0.0167	0.078	11.23
Cognitive Irritation	0.10	0.09	1.08	0.28		0.0338	*	0.0205	0.0509	9.33
Emotional Irritation	0.22	0.05	4.37	0.00	***	0.0865	*	0.0511	0.1321	23.89
Distraction (WRRQ)	0.01	0.07	0.17	0.86		0.0238		0.0124	0.0413	6.58
Inability to Recover (FABA)	0.26	0.09	2.81	0.01	**	0.0539	*	0.0351	0.076	14.88
Adjusted *R*-squared						0.3447				
*F* [10, 377]						20.83				
*p*						<0.001				

Note. *n* = 388. b = unstandardized regression coefficient; SE = standard error; RW = raw relative weight (within rounding error, raw weights will sum to R-squared); CI-L = lower bound of confidence interval used to test the statistical significance of raw weight; CI-U = upper bound of confidence interval used to test the statistical significance of raw weight; RS-RW (%) = relative weight rescaled as a percentage of predicted variance in the criterion variable attributed to each predictor (within rounding error, rescaled weights sum to 100%). Overcommitment (4 Items) = composite of items 2, 3, 4, and 5; REQ = recovery experience questionnaire; WRRQ = work-related rumination questionnaire; FABA = faulty attitude and behavior analysis. * *p* < 0.05. *** *p* < 0.001.

**Table 10 ijerph-20-03573-t010:** Coefficients of the relative weight analyses predicting emotional fatigue (Sample 2).

Criterion	Emotional Fatigue									
Predictor	*b*	*SE*	*T*	*p*		RW		CI-L	CI-U	RS-RW (%)
Intercept	1.12	0.56	1.99	0.05	*					
Overcommitment (4 Items)	−0.09	0.10	−0.92	0.36		0.0205		0.0116	0.0307	5.28
Psychological Detachment (REQ)	0.10	0.08	1.29	0.20		0.0106		0.0056	0.0153	3.30
Affective Rumination (WRRQ)	0.08	0.08	1.09	0.28		0.0463	*	0.0238	0.0768	14.16
Problem-solving Pondering (WRRQ)	−0.07	0.07	−0.99	0.32		0.0076		0.0039	0.0102	2.94
Positive Work Reflection	−0.10	0.05	−1.80	0.07	†	0.0227		0.0045	0.0507	7.52
Negative Work Reflection	0.07	0.06	1.21	0.23		0.0260		0.0087	0.0568	9.16
Cognitive Irritation	−0.03	0.11	−0.29	0.77		0.0245		0.0142	0.0369	7.12
Emotional Irritation	0.24	0.06	4.04	0.00	***	0.0779	*	0.0396	0.1251	27.52
Distraction (WRRQ)	−0.02	0.08	−0.28	0.78		0.0194		0.0096	0.0343	5.76
Inability to Recover (FABA)	0.46	0.11	4.31	0.00	***	0.0642	*	0.0399	0.0918	17.24
Adjusted *R*-squared						0.3013				
*F* [10, 377]						17.43				
*p*						<0.001				

Note. *n* = 388. b = unstandardized regression coefficient; SE = standard error; RW = raw relative weight (within rounding error, raw weights will sum to R-squared); CI-L = lower bound of confidence interval used to test the statistical significance of raw weight; CI-U = upper bound of confidence interval used to test the statistical significance of raw weight; RS-RW (%) = relative weight rescaled as a percentage of predicted variance in the criterion variable attributed to each predictor (within rounding error, rescaled weights sum to 100%). Overcommitment (4 Items) = composite of items 2, 3, 4, and 5; REQ = recovery experience questionnaire; WRRQ = work-related rumination questionnaire; FABA = faulty attitude and behavior analysis. † *p* < 0.10. * *p* < 0.05. *** *p* < 0.001.

**Table 11 ijerph-20-03573-t011:** Coefficients of the relative weight analyses predicting burnout (Sample 2).

Criterion	Burnout									
Predictor	*b*	*SE*	*T*	*p*		RW		CI-L	CI-U	RS-RW (%)
Intercept	1.86	0.49	3.76	0.00	***					
Overcommitment (4 Items)	−0.09	0.09	−0.99	0.32		0.0229	*	0.0131	0.0333	5.06
Psychological Detachment (REQ)	−0.08	0.07	−1.13	0.26		0.0217	*	0.01	0.0373	4.80
Affective Rumination (WRRQ)	0.19	0.07	2.87	0.00	**	0.0758		0.044	0.1108	16.77
Problem-Solving Pondering (WRRQ)	−0.04	0.06	−0.07	0.51		0.0088	*	0.0047	0.0131	1.94
Positive Work Reflection	−0.56	0.05	−3.42	0.00	***	0.0483	*	0.0195	0.0846	10.67
Negative Work Reflection	0.15	0.05	3.17	0.00	**	0.0519	*	0.0251	0.0899	11.48
Cognitive Irritation	−0.08	0.09	−0.83	0.41		0.0253	*	0.0157	0.0367	5.59
Emotional Irritation	0.30	0.05	5.79	0.00	***	0.1158	*	0.0705	0.1711	25.61
Distraction (WRRQ)	−0.04	0.07	−0.52	0.60		0.0251	*	0.0121	0.0456	5.55
Inability to Recover (FABA)	0.27	0.09	2.96	0.00	**	0.0566		0.0367	0.0784	12.52
Adjusted *R*-squared						0.4377				
*F* [10, 377]						31.28				
*p*						<0.001				

Note. *n* = 388. b = unstandardized regression coefficient; SE = standard error; RW = raw relative weight (within rounding error, raw weights will sum to R-squared); CI-L = lower bound of confidence interval used to test the statistical significance of raw weight; CI-U = upper bound of confidence interval used to test the statistical significance of raw weight; RS-RW (%) = relative weight rescaled as a percentage of predicted variance in the criterion variable attributed to each predictor (within rounding error, rescaled weights sum to 100%). Overcommitment (4 Items) = composite of items 2, 3, 4, and 5; REQ = recovery experience questionnaire; WRRQ = work-related rumination questionnaire; FABA = faulty attitude and behavior analysis. * *p* < 0.05. ** *p* < 0.01. *** *p* < 0.001.

**Table 12 ijerph-20-03573-t012:** Coefficients of the relative weight analyses predicting psychosomatic complaints (Sample 2).

Criterion	Psychosomatic Complaints									
Predictor	*b*	*SE*	*T*	*p*		RW		CI-L	CI-U	RS-RW (%)
Intercept	0.75	0.37	2.06	0.04	*					
Overcommitment (4 Items)	0.06	0.07	0.87	0.38		0.0224		0.0109	0.0378	8.77
Psychological Detachment (REQ)	0.02	0.05	0.46	0.65		0.0088		0.0037	0.0157	3.44
Affective Rumination (WRRQ)	0.15	0.05	3.07	0.00	**	0.0602	*	0.0322	0.0979	23.59
Problem-Solving Pondering (WRRQ)	−0.12	0.05	−2.46	0.01	*	0.0064		0.0027	0.012	2.52
Positive Work Reflection	0.03	0.04	0.97	0.34		0.0039		0.0013	0.0118	1.54
Negative Work Reflection	0.02	0.04	0.47	0.64		0.0153		0.0049	0.0352	5.99
Cognitive Irritation	−0.02	0.07	−0.34	0.73		0.0173		0.0088	0.0285	6.76
Emotional Irritation	0.15	0.04	3.85	0.00	***	0.0680	*	0.0324	0.119	26.64
Distraction (WRRQ)	0.01	0.05	−0.23	0.81		0.0130		0.0061	0.0241	5.08
Inability to Recover (FABA)	0.16	0.07	2.34	0.02	*	0.0400	*	0.0214	0.0635	15.67
*R*-squared						0.2357				
*F* [10, 377]						13.00				
*p*						<0.001				

Note. *n* = 388. b = unstandardized regression coefficient; SE = standard error; RW = raw relative weight (within rounding error, raw weights will sum to R-squared); CI-L = lower bound of confidence interval used to test the statistical significance of raw weight; CI-U = upper bound of confidence interval used to test the statistical significance of raw weight; RS-RW (%) = relative weight rescaled as a percentage of predicted variance in the criterion variable attributed to each predictor (within rounding error, rescaled weights sum to 100%). Overcommitment (4 Items) = composite of items 2, 3, 4, and 5; REQ = recovery experience questionnaire; WRRQ = work-related rumination questionnaire; FABA = faulty attitude and behavior analysis. * *p* < 0.05. ** *p* < 0.01. *** *p* < 0.001.

**Table 13 ijerph-20-03573-t013:** Coefficients of the relative weight analyses predicting satisfaction with life (Sample 2).

Criterion	Satisfaction with Life									
Predictor	*b*	*SE*	*T*	*p*		RW		CI-L	CI-U	RS-RW (%)
Intercept	2.68	0.70	3.82	0.00	***					
Overcommitment (4 Items)	0.45	0.13	3.57	0.00	***	0.0145		0.0054	0.0307	7.77
Psychological Detachment (REQ)	0.06	0.10	0.60	0.55		0.0053		0.0016	0.0086	2.85
Affective Rumination (WRRQ)	0.04	0.09	0.39	0.69		0.0102		0.0048	0.0218	5.46
Problem-Solving Pondering (WRRQ)	0.10	0.09	1.16	0.25		0.0078		0.0026	0.0235	4.17
Positive Work Reflection	0.27	0.07	4.03	0.00	***	0.0568	*	0.0237	0.1023	30.39
Negative Work Reflection	−0.03	0.07	−0.45	0.65		0.0055		0.0016	0.0204	2.96
Cognitive Irritation	0.12	0.13	0.89	0.38		0.0076		0.0027	0.0105	4.04
Emotional Irritation	−0.28	0.07	−3.83	0.00	***	0.0496		0.0201	0.0957	26.55
Distraction (WRRQ)	0.22	0.10	2.20	0.03	*	0.0124		0.0038	0.0332	6.66
Inability to Recover (FABA)	−0.35	0.13	−2.68	0.01	**	0.0171		0.007	0.0322	9.16
*R*-squared						0.1655				
*F* [10, 377]						8.72				
*p*						<0.001				

Note. *n* = 388. b = unstandardized regression coefficient; SE = standard error; RW = raw relative weight (within rounding error, raw weights will sum to R-squared); CI-L = lower bound of confidence interval used to test the statistical significance of raw weight; CI-U = upper bound of confidence interval used to test the statistical significance of raw weight; RS-RW (%) = relative weight rescaled as a percentage of predicted variance in the criterion variable attributed to each predictor (within rounding error, rescaled weights sum to 100%). Overcommitment (4 Items) = composite of items 2, 3, 4, and 5; REQ = recovery experience questionnaire; WRRQ = work-related rumination questionnaire; FABA = faulty attitude and behavior analysis. * *p* < 0.05. ** *p* < 0.01. *** *p* < 0.001.

## Data Availability

Correlation matrices are reported in the manuscript and the Appendix A. Raw and processed data supporting reported results can be accessed at https://osf.io/5tq6r/. Most recent materials uploaded on 3 January 2023.

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
