# Peer review of "Too Committed to Switch Off—Capturing and Organizing the Full Range of Work-Related Rumination from Detachment to Overcommitment"

_ijerph, 2023, doi:10.3390/ijerph20043573_

Round 1

Reviewer 1 Report

Dear Authors,

Please find attached my review report.

Best of Luck!

Reviewer 2 Report

Thank you for the opportunity to rate this article.

Overall, the article is well written and has great academic and practical contributions. However, some elements still cause me concern:

1- It is recommended to use a formal convergent analysis test and a formal test for discriminant analysis.

2- It is recommended to use McDonald's omega as a measure of reliability in addition to Cronbach's alpha.

3- It is not clear whether a tetrachoric or polychoric matrix was used to estimate the parameters in the Factor Analysis.

4- What do the three dimensions of Factor Analysis mean? What is the point of carrying out an analysis of this nature if the items are not grouped into 3 dimensions in the other analyses?

5- Items with low factor loading and commonality were not excluded after the factor analysis? What is the reason for not deleting?

6- Some methodological steps, such as the use of regression models, are not mentioned in the methodology. It is important that all procedures are clear from the reading of the methodology.

7- The article does not provide any evidence that the regression models are valid. For example, residual analysis.

8- R2 also has low values. Is this model really appropriate for relating the factors?

9- I believe that a theoretical model relating the factors could be more appropriate than the use of multiple regression.

The authors could use structural equation modeling to more appropriately model the relationship between the factors.

I recommend inserting such a limitation in the article and leaving it as a suggestion for future work. Authors may cite an article by Souza et al. (2021) that made use of factor analysis, regression model and structural equation model

Reference:

Souza, D.S.F. de; Silva, J.M.N. da; Santos, J.V. de O.; Alcântara, M.S.B.; Torres, M.G.L. Influence of Risk Factors Associated with Musculoskeletal Disorders on an Inner Population of Northeastern Brazil. Int J Ind Ergon 2021, 86, doi:10.1016/j.ergon.2021.103198.
